# Development of a Laser Gas Analyzer for Fast CO_2_ and H_2_O Flux Measurements Utilizing Derivative Absorption Spectroscopy at a 100 Hz Data Rate

**DOI:** 10.3390/s21103392

**Published:** 2021-05-13

**Authors:** Mingxing Li, Ruifeng Kan, Yabai He, Jianguo Liu, Zhenyu Xu, Bing Chen, Lu Yao, Jun Ruan, Huihui Xia, Hao Deng, Xueli Fan, Bangyi Tao, Xueling Cheng

**Affiliations:** 1AnHui Institute of Optics and Fine Mechanics, HeFei Institutes of Physical Sciences, Chinese Academy of Sciences, Hefei 230031, China; mxli@aiofm.ac.cn (M.L.); jgliu@aiofm.ac.cn (J.L.); zyxu@aiofm.ac.cn (Z.X.); bchen@aiofm.ac.cn (B.C.); lyao@aiofm.ac.cn (L.Y.); ruanjun@aiofm.ac.cn (J.R.); hhxia@aiofm.ac.cn (H.X.); hdeng@aiofm.ac.cn (H.D.); xlfan@aiofm.ac.cn (X.F.); 2University of Science and Technology of China, Hefei 230026, China; 3State Key Laboratory of Satellite Ocean Environment Dynamics, Second Institute of Oceanography, Ministry of Natural Resources, Hangzhou 310012, China; taobangyi@sio.org.cn; 4State Key Laboratory of Atmospheric Boundary Layer Physics and Atmospheric Chemistry, Institute of Atmospheric Physics, Chinese Academy of Sciences, Beijing 100029, China; chengxl@mail.iap.ac.cn

**Keywords:** laser gas analyzer, flux measurement, eddy covariance method, derivative absorption spectroscopy, gas sensors

## Abstract

We report the development of a laser gas analyzer that measures gas concentrations at a data rate of 100 Hz. This fast data rate helps eddy covariance calculations for gas fluxes in turbulent high wind speed environments. The laser gas analyzer is based on derivative laser absorption spectroscopy and set for measurements of water vapor (H_2_O, at wavelength ~1392 nm) and carbon dioxide (CO_2_, at ~2004 nm). This instrument, in combination with an ultrasonic anemometer, has been tested experimentally in both marine and terrestrial environments. First, we compared the accuracy of results between the laser gas analyzer and a high-quality commercial instrument with a max data rate of 20 Hz. We then analyzed and compared the correlation of H_2_O flux results at data rates of 100 Hz and 20 Hz in both high and low wind speeds to verify the contribution of high frequency components. The measurement results show that the contribution of 100 Hz data rate to flux calculations is about 11% compared to that measured with 20 Hz data rate, in an environment with wind speed of ~10 m/s. Therefore, it shows that the laser gas analyzer with high detection frequency is more suitable for measurements in high wind speed environments.

## 1. Introduction

The exchange of energy and mass between the ocean and atmosphere has significant impacts on the global environment, climate, and ecological balance. Flux measurements of heat, water, carbon dioxide, and methane, as well as other trace gases have been widely used to estimate the exchange of energy and mass [1,2,3,4,5]. With decades of technological development, the eddy covariance method has become a preferred method for direct flux estimations in turbulent motions without parametric assumptions, and is widely used in ecological flux observations [6].

Generally, the physical principle for the eddy covariance method is to measure the quantity of molecules moving upward or downward over time, and the speed in which they travel. Mathematically it can be represented as a covariance between measurements of vertical velocity of the upward or downward movements, and the concentration of the entity of interest [7]. The basic equipment for a flux measurement system mainly includes a three-dimensional ultrasonic anemometer and a gas analyzer. In the last decade, substantial progress has been made in the development of spectroscopic trace gas sensing technologies. This includes non-dispersive infrared spectroscopy (NDIR), tunable diode laser absorption spectroscopy (TDLAS), quantum cascade laser absorption spectroscopy (QCL-TDLAS), cavity ring-down spectroscopy (CRDS), and photoacoustic spectroscopy (PAS). Spectroscopic methods have advantages of high selectivity, high sensitivity, long-term stability, and have been applied for eddy covariance measurements. For example, Fortuniak et al. measured the greenhouse gases (CO_2_, CH_4_, H_2_O) at the wetlands of Biebrza National Park in Poland by using a sonic anemometer and gas analyzers (LI-COR LI-7500-H_2_O/CO_2_ and LI-7700-CH_4_) operating with 10 Hz frequency [8]. Kormann et al. developed a novel tunable diode laser absorption spectrometer for trace gas flux measurements based on micrometeorological techniques where the spectrometer was used to measure CH_4_ and N_2_O fluxes from rice paddies and tropical ecosystems [9]. Christian et al. tested a performance of a quantum cascade laser (QCL)-based N_2_O flux measurements against gas chromatography (GC) [10]. Crosson developed an analyzer based on cavity ring-down spectroscopy to measure the concentrations of CO_2_, H_2_O and CH_4_ [11]. He et al. developed a unique open-path CRDS technique for atmospheric sensing [12]; and Gong et al. recently developed a high-sensitivity resonant photoacoustic sensor for remote CH_4_ gas detection at ppb-levels [13,14].

Turbulent changes happen very quickly, and the respective changes are very small in concentration, density, or temperature. It is therefore necessary to use an instrument with high precision and fast data rate of measurements, especially in high wind environments. Nevertheless, data rates of flux measurements reported in literature are typically around 20 Hz or slower. The 20 Hz frequency detection may cause data loss and inaccuracy for analyzing the gas exchange and flux. For trace gases measurements, tunable laser absorption spectroscopy was developed decades ago as an ideal analysis and measurement technology, which has the advantages of high resolution, high selectivity, and high sensitivity [15,16]. It is widely used in the fields of greenhouse gas detection, toxic and hazardous gas detection in chemical parks, respiratory diagnosis, aero-engine combustion flow field diagnosis, deep-sea dissolved gas, and isotope detection [17,18,19,20,21,22,23,24].

In this work, we have developed a simple and compact laser gas analyzer with a data rate of 100 Hz, based on laser absorption spectroscopy and derivative absorption spectroscopy. The analyzer is designed by using two diode DFB lasers operating at wavelengths of ~2004 nm for CO_2_ and ~1392 nm for H_2_O measurements. Meanwhile, we have designed a multi-pass cell with a 20 m optical path length for CO_2_ absorption measurements and a single-path cell of 15 cm optical path length for H_2_O absorption measurements, as well as a miniaturized TDLAS electronics system. By developing a fast data processing of derivative absorption spectroscopy, we were able to achieve gas concentration measurements at a 100 Hz data rate. The system was tested in high and low wind speed environments by field measurements on an offshore platform in the Yellow Sea near the Yan-tai city in Shandong province and on the Jue-hua Island near Huludao city in Liaoning province. We compare the accuracy of results between our laser gas analyzer and a commercial instrument LI-COR LI-7500. Finally, we analyzed and compared the impact of data rates between 100 Hz and 20 Hz in high and low wind speeds to verify the contribution of high frequency detection.

## 2. Materials and Methods

### 2.1. Transmission-Intensity-Normalized Second-Derivative Spectroscopy

A direct tunable diode laser absorption spectroscopy (dTDLAS) is a reliable means for trace gas detections as it is relatively simple in construction, easy to handle, and reliable to use [25]. The technology is based on an attenuation of laser radiation due to absorption as descript by the Lambert–Beer’s law, which can be written as:(1)$$I(\nu )=I_{0}(\nu )\cdot \exp[-\varepsilon (\nu )\cdot L\cdot C],$$
where $I_{0}(\nu )$ is the incident intensity of the laser radiation of frequency ν. After passing through an absorbing medium, where optical path length is L and gas concentration is C, the transmitted intensity I(ν) is detected. The concentration-normalized absorption coefficient ε(ν) can be described by Equation (2):(2)ε(ν)=S(T)⋅P⋅ϕ(P,T,ν),
where P is the total pressure, S(T) is the temperature-dependent line strength, ϕ(P,T,ν) is the line shape function which is pressure and temperature dependent [26,27].

However, the detection limit of dTDLAS is affected by noise contributions in the measurement signal. The data analysis during concentration inversion also involves numerical division, logarithmic calculations, and possible nonlinear least-squares fitting. This type of calculation-intensive analysis is a challenge for the simple microcontrollers typically used in such measurement instruments, and slows the data acquisition rate. To improve on this, a derivative spectroscopy technique [28,29] can be applied. By processing spectral signal with second-order differential, the derivative spectral signal is obtained, and correlated with gas concentration. The transmission-intensity-normalized first and second derivatives of measurement signals can be written by Equations (3) and (4), respectively.
(3)$$\frac{dI}{d\nu }/I=\frac{dI_{0}}{d\nu }/I_{0}-L\cdot C\cdot \frac{d\varepsilon }{d\nu },$$
(4)$$\frac{d^{2}I}{d\nu^{2}}/I=\frac{d^{2}I_{0}}{d\nu^{2}}/I_{0}+{\left(L\cdot C\cdot \frac{d\varepsilon }{d\nu }\right)}^{2}-2\cdot L\cdot C\cdot \frac{d\varepsilon }{d\nu }\times \frac{dI_{0}}{d\nu }/I_{0}-L\cdot C\cdot \frac{d^{2}\varepsilon }{d\nu^{2}},$$

When a linearly ramp is used as drive current to a diode laser, the first term of Equation (4) is zero in an ideal case when changes of laser intensity are proportional to changes in its drive current. The residual deviation from zero is not dependent on the gas absorption and can be treated as an offset background. The values of the second and third term are zero at the center frequency of an absorption line, where the curvature slope (i.e., first derivative) is zero. The fourth term (second derivative) reaches a maximum value at the line center. Therefore, the transmission-intensity-normalized second derivative spectra have a linear relationship with the concentration of the absorbing medium.

### 2.2. Method of Flux Measurements

The eddy covariance (EC) method has been widely used for direct measurements of surface atmosphere exchange. It uses the covariance between vertical velocity $w_{i}$ in wind speed and fast variations of $C_{i}$ in trace concentration. The EC flux F can be calculated from a recorded time series of *N* measurements as:(5)$$F=\bar{C^{\prime }\cdot w^{\prime }}=\frac{1}{N}\sum_{i=1}^{N}\left[(C_{i}-\bar{C})\cdot (w_{i}-\bar{w})\right]=\frac{1}{N}\sum_{i=1}^{N}{C^{\prime }}_{i}\cdot {w^{\prime }}_{i},$$
where $C^{\prime }$ and $w^{\prime }$ are the instantaneous deviations from their corresponding mean values $\bar{C}$ and $\bar{w}$, respectively. For this work, an ultrasonic anemometer (GILL-HS100) was used to determine vertical wind speeds. The instantaneous gas concentrations of the species of interest (i.e., H_2_O vapor and CO_2_ gas) were measured with our newly developed laser gas analyzer at a fast data rate of 100 Hz. As eddies occur on a wide range of timescales, it is necessary to use sufficiently long averaging time for calculating mean values. For this study, a time interval of about 5 min was chosen for calculating the average value when operating at a data rate of 100 Hz (resulting in a total of 30,000 data points). Alternatively, the effective data sampling can be slowed to a lower rate (e.g., 20 Hz), or the averaging time base can be extended.

### 2.3. Selection of Spectral Absorption Lines

To achieve reliable measurements of trace gas concentrations, a spectral simulation is performed to determine whether the selected lines have sufficient strengths for measurements and are well isolated from the absorptions by other gas species without any serious interference. H_2_O and CO_2_ have several strong absorption bands in the infrared spectral range between 1.0 μm and 2.5 μm, as shown in Figure 1 [30]. For example, the line strengths of CO_2_ near 2.0 μm wavelength region are much stronger than in 1.6 μm region. So in this study, we used a ~2.0 μm diode laser for more sensitive detections. Figure 1c,d show the simulation of spectral absorption around 4991 cm^−1^ and 7181 cm^−1^, based on the HITRAN2016 database [31] for 2% H_2_O and 400 ppmv CO_2_ in air under nominal conditions (*P* = 1 atm, *T* = 296 K, path length *L* = 15 cm for H_2_O or 20 m for CO_2_, respectively). The results indicate that the target lines for H_2_O and CO_2_ detection are appropriate with minimum spectral inference. So the diode lasers of 1.392 μm wavelength (NEL, NLK1E5GAAA) and 2.004 μm wavelength (NEL, KELD1G5BAAA) are used in this work.

### 2.4. Experimental Setup

A schematic of the laser gas analyzer and experimental setup developed for flux measurement is shown in Figure 2, comprised of three units: a laser gas analyzer, an ultrasonic anemometer for three-dimensional wind speeds, power supply, and data acquisition. Among them, our developed laser gas analyzer consists of a miniaturized TDLAS measuring system and two gas absorption cells. The two diode lasers are driven by current controllers and temperature controllers with precision setting voltages, generated by digital-to-analog converters (DAC, Analog Devices AD5682, 14 bits) in combination with a microcontroller (ST, STM32H743VIT6). The wavelengths of diode lasers are ramped at a rate of 2 kHz via their operation currents. The fiber-coupled laser output is collimated and focused to either a single-pass absorption cell (15 cm optical path length) for H_2_O measurements, or a multi-pass absorption cell (Herriott style, 50 passes, total 20 m optical path length, 51-mm dimeter mirrors with 99.99% highly-reflective dielectric coatings around 2.0 μm) for CO_2_ measurements. The laser radiation is detected by a wavelength-extended (up-to 2.6 μm) InGaAs photodiode (GPD Optoelectronics), and then amplified (Analog Devices AD8065) and recorded by an analog-to-digital converter (ADC, 16 bits, on the STM32H743 microcontroller). The final results of gas concentrations and wind speeds are sent to a laptop computer by an Ethernet port and saved to a memory card. A GPS receiver is used to provide time information for synchronizing the data between the ultrasonic anemometer and the laser gas analyzer.

## 3. Results and Discussion

### 3.1. Signal Processing

To quickly process the measurement data for this project, we applied the Savitzky–Golay filter method [32] for signal smoothing and differentiation. This digital filtering technique fits successive sub-sets of adjacent data points with a low-degree polynomial by the method of linear least squares. As a result of this moving-window data smoothing process, it increased the precision of the data without distorting the signal tendency. Another important aspect of this Savitzky–Golay filtering technique is that it also obtains derivative information of the signal profile based on the fitted polynomials. For spectroscopy applications, the Savitzky–Golay filtering technique can help to reduce signal noise and identify structure components in complex spectra [33,34]. It enables us to achieve fast numerical data analysis of recorded measurement transmission absorption spectra for determining gas concentrations in gas sensing applications.

Figure 3a shows one example of a CO_2_ measurement spectrum and the associated 1st and 2nd derivatives obtained via a Savitzky–Golay filtering. The transmission spectrum was acquired for samples of ~500 ppmv CO_2_ by the ADC with 450 data points, and averaged 16 times in successive laser current scans. Simulations of the absorbance and 2nd differentiation spectrum at 1 ppmv CO_2_ concentration are displayed in Figure 3b. The 16 bits ADC for recording the photodetector signal has sufficient resolution (i.e., 1/65,535) to cover concentrations from ~2000 ppmv down to sub ppmv. Our noise-limited detection sensitivity corresponds to an absorbance level of ~1.5 × 10^−^^4^.

### 3.2. Gas Analyzer Performance

To evaluate the performance of numerical analysis of derivative absorption spectroscopy, we prepared a series of reference gas mixtures of CO_2_ and H_2_O with different concentrations and made measurements at atmospheric pressure. The subsequent second derivative absorption signals were calculated by using the Savitzy–Golay filter method, as shown in Figure 4a for CO_2_ and Figure 4c for H_2_O. As expected, a linear correlation between the signal peak magnitude of the second derivative spectra and the CO_2_ and H_2_O concentrations were confirmed. The results presented in Figure 4b,d show a good linear dependence (adj. R^2^ = 0.995 for CO_2_ and adj. R^2^ = 0.999 for H_2_O), and demonstrate that the algorithm is valid for trace gas measurements. The slope of the fitted straight-line also serves as a conversion coefficient between the experimental measurements and the resulting gas concentrations.

The detection limit of the developed laser gas analyzer is evaluated by using Allan variance plots [35]. Figure 5 shows the results of Allan deviations for CO_2_ and H_2_O concentration measurements with a sample containing 500 ppmv CO_2_ and 2% H_2_O, plotted in a log–log scale. The measurement noise at 100 Hz data rate (i.e., 0.01 s averaging time) is about 0.40 ppmv for CO_2,_ and 8.17 ppmv for H_2_O, respectively. As the averaging time increases, the minimum reaches about 0.026 ppmv for CO_2_ at 6 s integration time, and 3.12 ppmv for H_2_O at 0.13 s integration time. Table 1 summarizes the performance comparison between the laser gas analyzer we developed, and the commercial instrument LI-7500-CO_2_/H_2_O based on non-dispersive infrared technology [36]. The TDLAS gas analyzer we developed performs slightly better for H_2_O than the LI-7500 instrument, but was slightly worse for CO_2_. The major advancement of our device is its fast maximum measurement data rate of 100 Hz, corresponding to a time resolution of 10 ms, which would enable observation of fast turbulent motions for eddy covariance.

### 3.3. Experimental Field Measurements

In order to test the performance of the developed gas analyzer for H_2_O and CO_2_ fluxes measurements under low and high wind speed environments, we conducted the field measurements at two distinctive sites, as depicted in Figure 6. Figure 6a shows the installation of our integrated instruments and an ultrasonic anemometer on an offshore platform in the Yellow Sea of Yan-tai city in Shandong province (Site-A) with high-wind-speed marine environment. Figure 6b shows the installation on the Jue-hua Island of Huludao city in Liaoning province (Site-B) with low-wind-speed terrestrial environment. The commercial LI-7500-CO_2_/H_2_O instrument is also installed nearby to calibrate and compare the accuracy of measurements.

The results of comparison measurements for an hour in the marine environment (Figure 6a) with wind speed of about 13 m/s are displayed in Figure 7a,b, which show CO_2_ and H_2_O concentrations determined by our laser gas analyzer and the commercial LI-7500 analyzer. Our H_2_O sensing unit was installed close to the LI-7500, while the CO_2_ sensing unit was slightly further away. The H_2_O concentration measurements show good agreement between the two instruments. The difference of CO_2_ concentration trend between our laser gas analyzer and the LI-7500 analyzer was partly due to its installation location, which is slightly further away, therefore had different wind conditions. Another contributing factor might be the mounting and housing of the multi-pass sensing unit, which we will investigate in future studies. Furthermore, we measured CO_2_ absorption in the wavelength region of its ~2.0 μm absorption band, while the LI-7500 analyzer operated around the ~4.2 μm stronger absorption band, resulting in a higher sensitivity with shorter optical path length. The detected concentration of CO_2_ for both analyzers in the marine environment ranges from 380 ppmv to 410 ppmv, and the detected concentration of H_2_O ranges for 22 mmol/mol to 25 mmol/mol.

For the investigation of the frequency characteristics of the measurements and the time response of the instruments, we applied the frequency power density method to analyze the recorded time series of the concentration data [37]. Figure 8a,b show the comparison of power density spectra for CO_2_ and H_2_O concentrations between the laser gas analyzer and LI-7500 analyzer via a fast Fourier transform method. The slope of spectra (ploted in log–log scales) is approximate to −5/3 of the Kolmogorov theory in the inertial subrange [38], indicating that the laser gas analyzer is capable of measuring turbulence fluxes of CO_2_ and H_2_O via EC method.

### 3.4. Comparison of Flux Measurements under Low and High Wind Speeds

Measurements of wind speed form an integral part in the determination of flux of gas emissions and movements. Three-dimensional wind speeds are measured via an ultrasonic anemometer. However, the mounting of the anemometer may not be exactly vertical or horizontal to the ground. A coordinate rotation is needed to transform the measurement values of the anemometer to the three velocity components u, ν, w in respect to the ground, where w is the vertical wind velocity and is expected to have a mean value of zero. The components u and ν are the two wind velocities in the horizontal plane. The horizontal component ν is also aligned during the coordinate rotation to have a mean value of zero. Therefore, the component u represents the speed along horizontal wind direction. The results of wind speeds after rotating and H_2_O concentrations are plotted in Figure 9. The maximum horizontal wind speed in the marine environment of Site-A is about 13 m/s, whereas the maximum vertical wind speed is about 4 m/s in the terrestrial environment of Site-B.

Fast measurements can capture more details of rapid small-scale turbulence in air movements, especially in a high wind speed environment. Subsequently, this is expected to be an advantage for flux determination. We numerically analyzed the impact of data sampling rate by block averaging every 5 data samples of the original 100 Hz data set to obtain a 20 Hz data set. This 20 Hz data set loses frequency contributions above 20 Hz. Both data sets are computed for flux using a 5 min time base (see Equation (5)). The results of H_2_O fluxes are shown in Figure 10, for both high and low wind speeds.

As described in Figure 10a,b, the H_2_O fluxes computed for data rates of 20 Hz and 100 Hz can differ by up to 16% (adjustable R^2^ = 0.84) in the 10 m/s wind speed environment. As a comparison, the difference of H_2_O fluxes is about 5% (adjustable R^2^ = 0.95) in the 4 m/s wind speed environment, shown in Figure 10c,d. The difference suggests that the contribution of 100 Hz flux measurements increases by about 11%, as wind speed changes from 4 m/s to 10 m/s. The data points spread out further away from the straight-line in Figure 10b than in Figure 10d.

## 4. Conclusions

In this paper, a laser gas analyzer based on a second derivative laser absorption spectroscopy method has been developed to achieve a 100 Hz fast data rate, which is faster than that of a well-established 20 Hz commercial instrument. In combination with an ultrasonic anemometer, we applied the new laser gas analyzer for measurements of gas fluxes by using the eddy covariance method. Two DFB lasers operating at ~2.004 μm for CO_2_ and ~1.392 μm for H_2_O were used as optical sources. We built a multi-pass absorption cell of 20 m optical path length for CO_2_ measurements, and a single-pass absorption cell of 15 cm optical path length for H_2_O detection. A miniaturized TDLAS electronics system was designed for the operation and analysis of the gas concentration measurements. The gas analyzer achieves a detection limit of 8.17 ppmv for H_2_O and 0.40 ppmv for CO_2_ at 0.01 s integration time. Meanwhile, we made field measurements by installing the integrated instruments in two different environments to verify the influence of different wind speeds on flux measurements against a commercial instrument LI-7500. In general, the 100-Hz gas analyzer we developed has a wide prospect for flux measurement applications, especially when rapid turbulence is involved.

## Figures and Tables

**Figure 1 sensors-21-03392-f001:**
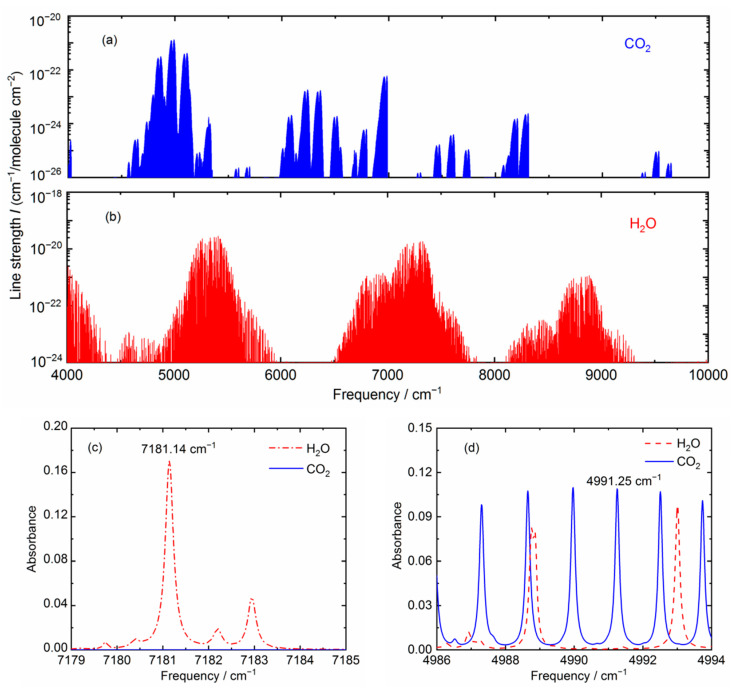
(**a**,**b**) Spectral line strengths; (**c**,**d**) simulation of spectral absorption with 2% H_2_O and 400 ppmv CO_2_ in air at a temperature of 296 K and pressure of 1 atm, in the near infrared wavelength regions (**c**) 1.392 μm, path length 15 cm, and (**d**) 2.004 μm, path length 20 m, respectively.

**Figure 2 sensors-21-03392-f002:**
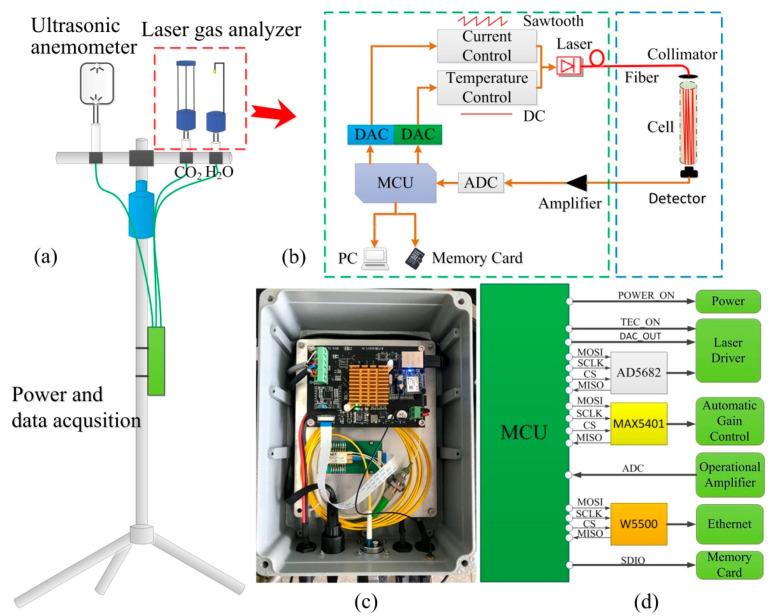
Schematic of the laser gas analyzer and experimental setup for flux measurements. (**a**) the three major components of the system; (**b**–**d**) details of the miniaturized TDLAS system.

**Figure 3 sensors-21-03392-f003:**
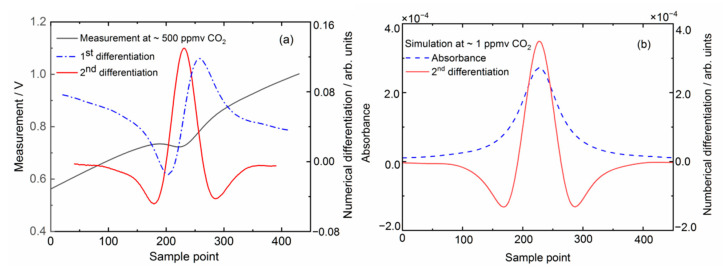
(**a**) A measurement example of tunable diode laser transmission spectrum of CO_2_ absorption, and the subsequent numerical analysis of its transmission-intensity-normalized 1st and 2nd differentiations by Savitzky–Golay filtering; (**b**) simulations of absorbance and 2nd differentiation at 1 ppmv CO_2_ concentration.

**Figure 4 sensors-21-03392-f004:**
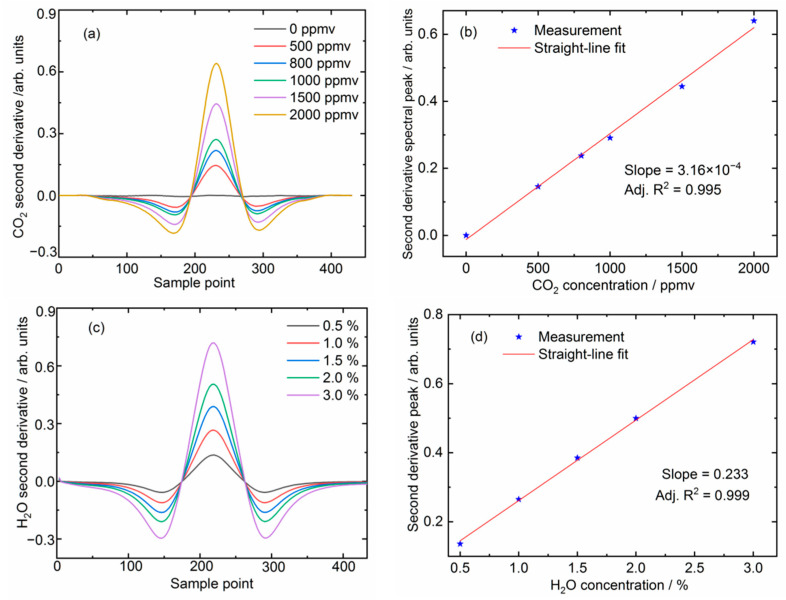
Numerically processed second derivative spectra of measurements at different gas concentrations of (**a**) CO_2_ and (**c**) H_2_O. The corresponding straight-line fits of the spectral peak magnitudes to the gas concentrations are displayed in (**b**,**d**).

**Figure 5 sensors-21-03392-f005:**
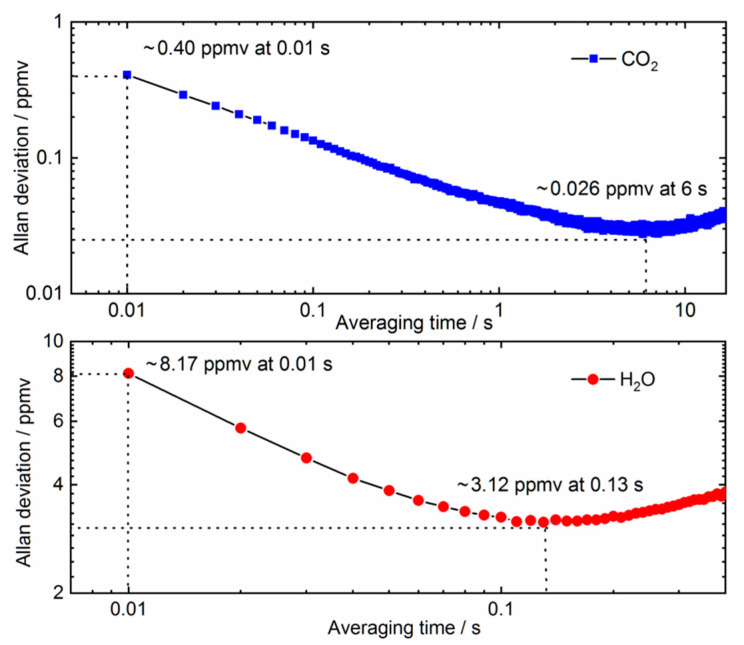
Allan deviation analyses of CO_2_ and H_2_O detection limits as a function of averaging time.

**Figure 6 sensors-21-03392-f006:**
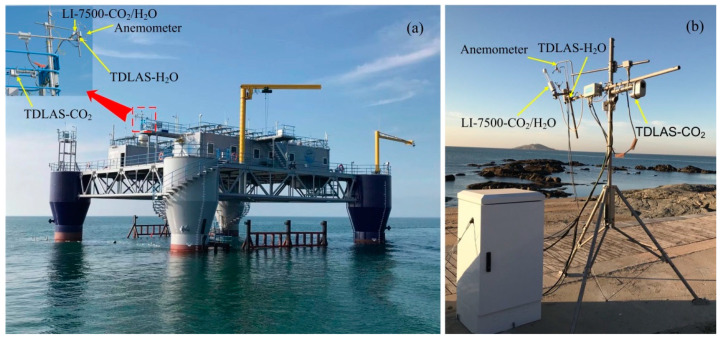
Photographs of the installation for field measurements at two sites of different environmental conditions. (**a**) Site-A of marine environment. (**b**) Site-B of terrestrial environment.

**Figure 7 sensors-21-03392-f007:**
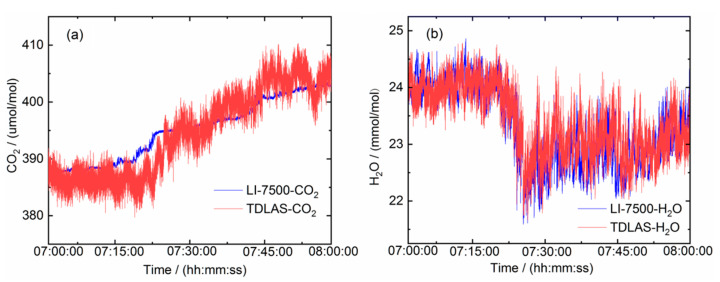
Comparison measurements between the laser gas analyzer and an LI-7500-CO_2_/H_2_O for an hour. (**a**) CO_2_ and (**b**) H_2_O measurements. The data rate of the TDLAS was at 100 Hz (with only one-fifth of the data points plotted in the graphs), whereas the LI-7500 was at its max 20 Hz.

**Figure 8 sensors-21-03392-f008:**
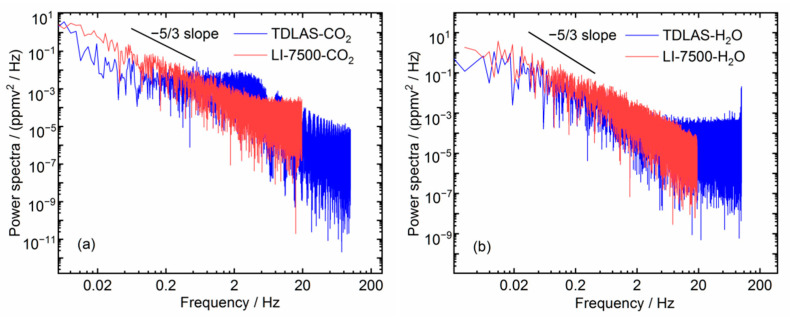
Power density analysis of concentration measurements between laser gas analyzer and LI-7500 analyzer for (**a**) CO_2_ and (**b**) H_2_O concentration measurements.

**Figure 9 sensors-21-03392-f009:**
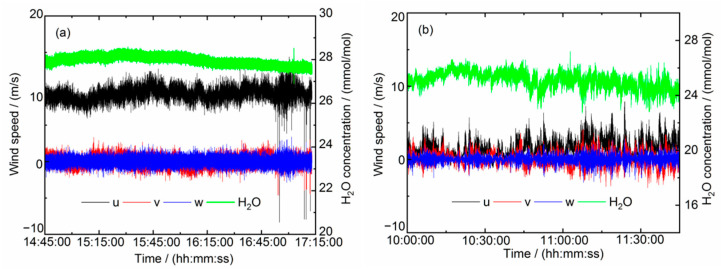
Measurement data of H_2_O concentrations and three-dimensional wind velocities (*u*, *v* for the two horizontal components, *w* for vertical). (**a**) Data of Site-A. (**b**) Data of Site-B.

**Figure 10 sensors-21-03392-f010:**
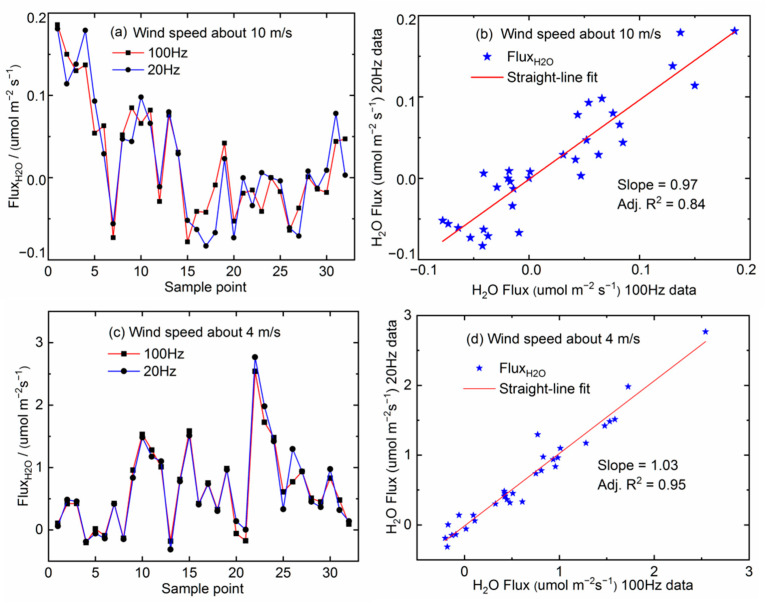
Comparison of H_2_O fluxes between 20 Hz and 100 Hz. (**a**) The comparison of H_2_O fluxes between 20 Hz and 100 Hz in 10 m/s wind speed environment. (**b**) The correlation analysis of H_2_O fluxes between 20 Hz and 100 Hz in 10 m/s wind speed environment. (**c**) The comparison of H_2_O fluxes between 20 Hz and 100 Hz in 4 m/s wind speed environment. (**d**) The correlation analysis of H_2_O fluxes between 20 Hz and 100 Hz in 4 m/s wind speed environment.

**Table 1 sensors-21-03392-t001:** Performance comparison between our laser gas analyzer and a commercial instrument.

	Our TDLAS Analyzer	LI-7500 Analyzer
Method	Laser absorption spectroscopy	Non-dispersive infrared spectroscopy
Detection limit, H_2_O	3.25 ppmv at 10 Hz 8.17 ppmv at 100 Hz	4.70 ppmv at 10 Hz 6.70 ppmv at 20 Hz
Detection limit, CO_2_	0.13 ppmv at 10 Hz 0.40 ppmv at 100 Hz	0.11 ppmv at 10 Hz 0.16 ppmv at 20 Hz
Maximum data rate	100 Hz	20 Hz

## Data Availability

All data will be made available on request to the correspondent author’s email with appropriate justification.

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
