# Peer review of "Development of a Laser Gas Analyzer for Fast CO2 and H2O Flux Measurements Utilizing Derivative Absorption Spectroscopy at a 100 Hz Data Rate"

_sensors, 2021, doi:10.3390/s21103392_

Round 1

Reviewer 1 Report

The authors MingXing Li et al. present a laser gas analyzer suitable for fast CO2 and H2O flux measurements utilizing derivative absorption spectroscopy aimed at high wind speed environments.

Questions to be addressed for a successful publication:

2.1. Transmission-intensity-normalized second-derivative spectroscopy

The authors present an interesting aspect of simplification for the generation of second-derivative transmission signals by using a Savitzky-Golay filtering technique:

  • Please elaborate the computational load difference to the claimed “calculation-intense” analysis by conventional direct absorption spectroscopy. Is a Savitzky-Golay filtering technique computationally less expensive than logarithm and inversion?
  • In the 2.4 experimental section a laptop is mentioned: is the data calculation (SG-filtering) applied offline?
  • Why is 2f-Wavelength Modulation Spectroscopy (2f-WMS) not taken into consideration? Please elaborate the decision.

2.4. Experimental Setup

  • More details to the 20m cell! Herriott configuration or White cell? Mirror material?

3.2. Gas analyzer performance

  • 4. 500 ppmv seems to be the lowest CO2 concentration step for the calibration results for a limit of detection of 0.4 ppmv! Please show a 2nd order spectrum for 1ppmv CO2. Signal to noise ratio for a 14bit ADC seems very limited.
  • Please remember the “lever” effect for very high CO2 concentrations (2000ppmv) in the case of Fig.4b calibration results. Very high concentrations speak for the linearity the sensor, though.
  • H2O calibration results/graphs missing…
  • Fig5: please state the concentration for the recorded Allan-Werle plot!

Author Response

Dear Reviewer,

Thank you for the comments and suggestions. We have carefully revised our manuscript based on these suggestions. Our point-by-point responses are included below.

Questions to be addressed for a successful publication:

2.1. Transmission-intensity-normalized second-derivative spectroscopy

The authors present an interesting aspect of simplification for the generation of second-derivative transmission signals by using a Savitzky-Golay filtering technique:

  • Please elaborate the computational load difference to the claimed “calculation-intense” analysis by conventional direct absorption spectroscopy. Is a Savitzky-Golay filtering technique computationally less expensive than logarithm and inversion?

Response:

The conventional direct absorption spectroscopy involves measurements of transmission laser beam intensity relative to absorption-free intensity as a function of scanned laser wavelength, logarithmic calculations, and fitting of measurement spectral profiles to model spectral profiles. The Levenberg-Marquardt method is widely used for this nonlinear fitting task, but it involves large amount of calculations.

The Savitzky-Golay filtering technique deals with only simple numerical calculations for obtaining derivative information of spectral profiles.

The following book contains comprehensive details on many topics of scientific numerical analyses:

Press, W.H.; Teukolsky, S.A.; Vetterling, W.T.; Flannery, B.P. Numerical recipes in Fortran 77: the art of scientific computing, 2nd ed., ISBN 0-521-43064-X; Cambridge University Press: Cambridge, UK, 1992; pp 644-649.

   Chapter 14.8 Savitzky_Golay Smoothing Filters

   Chapter 15. Modeling of Data

We have now added this book as a reference to the manuscript for technical details.

  • In the 2.4 experimental section a laptop is mentioned: is the data calculation (SG-filtering) applied offline?

Response:

The digital SG-filtering and gas concentration determination are performed by the microcontroller of the laser gas analyzer. A laptop is used for receiving the data of gas concentration and wind speed info, and for calculating the flux results.

We have revised the sentence to make it clearer: “The final results of gas concentrations and wind speeds are sent to a laptop computer…”

  • Why is 2f-Wavelength Modulation Spectroscopy (2f-WMS) not taken into consideration? Please elaborate the decision.

Response:

2f-Wavelength Modulation Spectroscopy Technology (2f-WMS) usually employs a Lock-In detection scheme for the extraction of the 2f information. The Lock-In detection/amplifier requires longer signal averaging. The output of the Lock-In amplifier needs further numerically processing.

In this research work, the primary purpose is to develop a compact simple system for robust concentration measurements at 100-Hz data rate. The second-derivative spectroscopy method is simple to implement and good in performance.

Alternatively in another work by our research group, we developed a scanned-wavelength modulation spectroscopy in combination with a fast 1f-phase detection method.

Li et al. “Compact open-path sensor for fast measurements of CO2 and H2O using scanned-wavelength modulation spectroscopy with 1f-phase method”, Sensors 20(7), 1910 (2020).

We have now added this paper as a reference to the manuscript.

2.4. Experimental Setup

  • More details to the 20m cell! Herriott configuration or White cell? Mirror material?

Response:

We have revised the manuscript by adding further details about the multi-pass cell: (Herriott style, 50 passes, total 20 m optical path length, 51-mm dimeter mirrors with 99.99% highly-reflective dielectric coatings around 2.0 mm)

3.2. Gas analyzer performance

  • Fig 4. 500 ppmv seems to be the lowest CO2 concentration step for the calibration results for a limit of detection of 0.4 ppmv! Please show a 2ndorder spectrum for 1ppmv CO2. Signal to noise ratio for a 14bit ADC seems very limited.

Response:

The ADC used in this work for recording the photodetector signal has a 16 bits resolution (i.e. 1 in 65535). At 1 ppmv CO2, the absorbance is about 2.7E-4. Therefore, the ADC has sufficient resolution to detect such a small relative change in transmission laser intensity.

We have now added simulations of absorbance and 2nd derivative spectrum of 1 ppmv CO2 to Figure 3 (Fig. 3a for 500 ppmv CO2, Fig. 3b for 1 ppmv CO2).

  • Please remember the “lever” effect for very high CO2 concentrations (2000ppmv) in the case of Fig.4b calibration results. Very high concentrations speak for the linearity the sensor, though.

Response:

We used variety concentrations between 0 and 2000 ppmv to evaluate the linearity performance by a straight-line fitting (Figure 4b & 4d). We found that the response of the gas analyzer does not have significant nonlinearity. A polynomial model could be considered but is not needed here.

  • H2O calibration results/graphs missing…

Response:

A series of reference gas mixture of H2O with different concentrations between 0.5% and 3% were used for calibrating the analyzer’s H2O measurements.

We have now added these H2O calibration results to Figure 4. (4a & 4b for CO2, 4c & 4d for H2O)

  • Fig5: please state the concentration for the recorded Allan-Werle plot!

Response:

We have revised the text in the manuscript: “…for CO2 and H2O concentration measurements with a sample containing 500 ppmv CO2 and 2% H2O,…”

Best regards,

MingXing Li

Reviewer 2 Report

The paper developed a laser gas analyzer based on a second derivative laser absorption spectroscopy method. And made a full comparison study between the performances of the established system and a commercial instrument. It is well written and the information on the principal and performances are in detail.

Author Response

Dear Reviewer,

Thank you for reviewing our manuscript. 
We are pleased with these supportive comments on our research and the manuscript.
We have made some further improvements to the revised manuscript. 

Best regards,
Mingxing Li

Reviewer 3 Report

The authors demonstrated a laser gas analyzer working in high wind speed environment. The structure is novel. The experimental results indicate that it is better than commercial gas analyzer LI-7500. Overall the paper is well written and good organized, but related certain points is better to be explained in more details. Following modifications are required:

  1. In figure 1, there is a CO2 absorption peak of CO2 on the right side of 4991.25, and it is further away from the absorption peak of H2O. Why choose 4991.25?
  2. Page5, line 165, “….The fiber-coupled laser output is collimated and focused to either a single-pass absorption cell (15 cm optical path length) for H2O measurements, or a multi-pass absorption cell (20 m optical path length) for CO2 measurements.” Why not use the same absorption cell?
  3. Page 8, line 243, “….while the CO2 sensing unit was slightly further away.”, It's better to put them in a similar place.
  4. In figure 7,according to the above, the performance of gas analyzer is different at different wind speeds. So at what wind speed is this measured?
  5. In the part of introduction, “In the last decade, sub-stantial progress has been made in the developments of spectroscopic trace gas sensing technologies, such as non-dispersive infrared spectroscopy (NDIR), tunable diode laser absorption spectroscopy (TDLAS), quantum cascade laser absorption spectroscopy (QCL-TDLAS), and cavity ring-down spectroscopy (CRDS)”, except for these spectroscopy techniques, the photoacoustic spectroscopy (PAS) is another effective spectroscopic trace gas sensing technology. Some latest references should be cited. A) Photoacoustics, 2021, 21: 100216. B) Optics Express, 2021, 29(9): 13600-13609.

Author Response

Dear Reviewer,

Thank you for the comments and suggestions. We have carefully revised our manuscript based on these suggestions. Our point-by-point responses are included below.

The authors demonstrated a laser gas analyzer working in high wind speed environment. The structure is novel. The experimental results indicate that it is better than commercial gas analyzer LI-7500. Overall the paper is well written and good organized, but related certain points is better to be explained in more details.

Response:

We are pleased with these supportive comments on our research and the manuscript.

Following modifications are required:

  1. In figure 1, there is a CO2 absorption peak of CO2 on the right side of 4991.25, and it is further away from the absorption peak of H2O Why choose 4991.25?

Response:

There are several CO2 and H2O absorption peaks around 4991 cm-1. On the right side of 4991.25 cm-1, there is another CO2 absorption peak at 4992.50 cm-1. However, it is seriously affected by another H2O absorption at 4991.94 cm-1. Therefore, the 4991.25 cm-1 CO2 absorption peak is least affected and chosen for use in this work.

We have revised the wavelength range in Figure 1d to show both H2O absorption lines.

  1. Page5, line 165, “….The fiber-coupled laser output is collimated and focused to either a single-pass absorption cell (15 cm optical path length) for H2O measurements, or a multi-pass absorption cell (20 m optical path length) for CO” Why not use the same absorption cell?

Response:

As the absorption line strengths and target concentrations of CO2 and H2O are much different, a short path length will be sufficient for H2O detection while a long path length will be needed for detecting weak CO2 absorptions. Furthermore, using the same absorption cell will require combining/separating laser beams from two different laser sources and make the setup more complex.

During this work, we found it was simpler and more flexible to use two separate cells. In another work by our research group, we have combined a single-pass cell and a multi-pass cell into a unified structure (Sensors 20, 1910 (2020)).

  1. Page 8, line 243, “….while the CO2 sensing unit was slightly further away.”, It's better to put them in a similar place.

Response:

Yes, it would be better to keep the separation between the two instruments as close as possible without causing significant disturbances to each other. In any future work, we will certainly want to make improvements in this aspect.

  1. In figure 7, according to the above, the performance of gas analyzer is different at different wind speeds. So at what wind speed is this measured?

Response:

The results displayed in Figure 7 were measured in a marine environment at a wind speed of about 13 m/s.

We have now added a statement about the wind speed in the manuscript.

  1. In the part of introduction, “In the last decade, substantial progress has been made in the developments of spectroscopic trace gas sensing technologies, such as non-dispersive infrared spectroscopy (NDIR), tunable diode laser absorption spectroscopy (TDLAS), quantum cascade laser absorption spectroscopy (QCL-TDLAS), and cavity ring-down spectroscopy (CRDS)”, except for these spectroscopy techniques, the photoacoustic spectroscopy (PAS) is another effective spectroscopic trace gas sensing technology. Some latest references should be cited. A) Photoacoustics, 2021, 21: 100216. B) Optics Express, 2021, 29(9): 13600-13609.

Response:

Thank you for the suggestion. Indeed, photoacoustic spectroscopy is a very effective spectroscopy technique and can achieve high detection sensitivity.

We have now added the PAS technique in the introduction and cited both the references in the manuscript.

Best regards,

MingXing Li

Round 2

Reviewer 2 Report

the munuscript is impoved. it can be accepted in present form.